# Design and Development of a Low-Cost UGV 3D Phenotyping Platform with Integrated LiDAR and Electric Slide Rail

**DOI:** 10.3390/plants12030483

**Published:** 2023-01-20

**Authors:** Shuangze Cai, Wenbo Gou, Weiliang Wen, Xianju Lu, Jiangchuan Fan, Xinyu Guo

**Affiliations:** 1School of Agricultural Equipment Engineering, Jiangsu University, Zhenjiang 212013, China; 2Beijing Key Lab of Digital Plant, National Engineering Research Center for Information Technology in Agriculture, Beijing 100097, China; 3Information Technology Research Center, Beijing Academy of Agriculture and Forestry Sciences, Beijing 100097, China; 4Beijing PAIDE Science and Technology Development Co., Ltd., Beijing 100097, China

**Keywords:** 3D phenotyping platform, electric slide rail, LiDAR, low-cost UGV, point cloud processing

## Abstract

Unmanned ground vehicles (UGV) have attracted much attention in crop phenotype monitoring due to their lightweight and flexibility. This paper describes a new UGV equipped with an electric slide rail and point cloud high-throughput acquisition and phenotype extraction system. The designed UGV is equipped with an autopilot system, a small electric slide rail, and Light Detection and Ranging (LiDAR) to achieve high-throughput, high-precision automatic crop point cloud acquisition and map building. The phenotype analysis system realized single plant segmentation and pipeline extraction of plant height and maximum crown width of the crop point cloud using the Random sampling consistency (RANSAC), Euclidean clustering, and k-means clustering algorithm. This phenotyping system was used to collect point cloud data and extract plant height and maximum crown width for 54 greenhouse-potted lettuce plants. The results showed that the correlation coefficient (R2) between the collected data and manual measurements were 0.97996 and 0.90975, respectively, while the root mean square error (RMSE) was 1.51 cm and 4.99 cm, respectively. At less than a tenth of the cost of the PlantEye F500, UGV achieves phenotypic data acquisition with less error and detects morphological trait differences between lettuce types. Thus, it could be suitable for actual 3D phenotypic measurements of greenhouse crops.

## 1. Introduction

Plant phenotypes are recognizable physical, physiological, and biochemical characteristics and traits that arise in part or whole due to the interaction of genes with the environment [1,2]. Plant phenomics has gradually become a key research area in basic and applied plant science in the past two decades. Nevertheless, inadequate phenotype detection technology is one of the major bottlenecks in crop breeding [3]. Traditional phenotype detection methods are labor-intensive, time-consuming, and subjective, with no uniform standards [4]. Therefore, machines that can replace manual plant phenotype detection are needed. Many crop phenotype information acquisition platforms have been recently developed in China and abroad. Based on the growth scenarios of crops, the platforms can be divided into field phenotyping and indoor phenotyping platforms [5]. Indoor phenotyping ones include platforms such as Crop 3D developed by the Institute of Botany, Chinese Academy of Sciences, and Scanalyzer 3D developed by LemnaTec, Germany, and the outdoor phenotyping platforms include Field Scanalyzer from Rothamsted Research Center, UK, and LQ-Fieldpheno from Agricultural Information Technology Research Center, Beijing, China. The efficiency and quality of phenotype information acquisition have been greatly improved due to the rapid development of sensor technology and equipment for plant phenotypes [6]. As a result, breeding programs focused on feeding billions of people worldwide have significantly improved [7].

The major phenotype acquisition devices include orbital [8], robotic [9], vehicle-mounted [10], suspension [11], and unmanned aerial vehicle-mounted [12]. However, these phenotyping devices, including vehicle-mounted and robotic phenotyping devices, can easily interfere with or even crush crops due to the complex outdoor conditions. As a result, their use is often characterized by a large amount of data, low ground resolution, much additional information (geographic location, light, temperature, water, air, and other environmental factors), non-uniform acquisition standards, high data uncertainty, low repeatability, and high timeliness [13]. The indoor phenotyping platform can simulate diverse crop growth conditions when combined with environmental control equipment (controlled greenhouse and artificial climate chamber) for the assessment of phenotypic plasticity and stability, identification of key phenotypic traits (yield, quality, and various resistance indicators) in all aspects, and obtaining statistically significant research conclusions. Therefore, they can achieve precise regulation, graded simulation, and automated precision collection, which cannot be easily achieved in outdoor environments [14]. Therefore, indoor phenotypic monitoring technologies are suitable for accurate and graded simulation and targeted crop growth and development research under complex experimental conditions [15].

Nevertheless, machine vision methods can be more accurate and effective in measuring key growth parameters (plant height and maximum crown width) of common crops, especially leafy vegetables [16]. However, the special structure of plants and the complex environment limit the high precision of plant phenotypic parameters through 2D images [17]. Therefore, the 3D structure is crucial for assessing plant growth status. Furthermore, studies have shown that equipment and technology for acquiring 3D phenotypes of crop canopies are crucial for quality breeding, scientific cultivation, and fine management [18,19]. Various devices and methods have been used to assess crop 3D information based on the principles of binocular stereo vision, structured light, Time of Flight (ToF), and multi-view stereo reconstruction (MVS). For example, Song et al. [20] obtained 3D point clouds of horticultural crops and achieved surface reconstruction based on binocular stereo vision. Hui et al. [21] reconstructed 3D plant point cloud models of cucumber, pepper, and eggplant and calculated phenotypic parameters, such as leaf length, leaf width, leaf area, plant height, and maximum canopy width based on multi-view stereo vision method and laser scanning method. LiDAR is widely used to acquire crop phenotypic information due to its high accuracy and fast scanning speed. Zheng et al. [22] also obtained 3D canopy point cloud data of trees and estimated their leaf area index using ground-based laser scanning. Sun et al. [23] extracted individual tree crowns and canopy width from aerial laser scanning data. Zhang et al. [24] analyzed the dynamic phenotypic changes of trees under wind disturbance using LiDAR.

In summary, UGVs equipped with LiDAR systems have received much attention due to their flexibility in monitoring crop phenotypes and high accuracy in the 3D reconstruction of crops. For example, UGVs equipped with LiDAR systems have been used to measure crop nitrogen status [25], estimate above-ground biomass [26], and measure planting density [27]. The commonly used orbital overhead travel phenotyping platform is limited by its high cost and immobility. Moreover, it lacks accuracy during sensor acquisition due to the vibration caused by the uneven ground when the UGV is moving. In this study, a new UGV phenotype platform equipped with an electric slide rail and phenotype data acquisition-analysis pipeline was developed using electric slide rail and LiDAR for accurate data acquisition. Furthermore, the 3D reconstruction and growth parameter measurement of lettuce grown in greenhouse pots were conducted. This study aimed to establish a low-cost automated crop growth non-destructive measurement system with good accuracy and practicality.

## 2. Methods

### 2.1. System Overview

The UGV-LiDAR phenotyping system consists of three main parts: the hardware part, the data acquisition part, and the data processing part (Figure 1): (1) The phenotype data acquisition hardware equipment contains a four-wheel drive self-propelled UGV, electric slide rail, LiDAR, real time kinematic global positioning system (RTK-GPS), and industrial computer; (2) the data acquisition control module includes the UGV navigation path setting, the electric slide rail’s moving speed and moving range setting, LiDAR start and stop, data saving path and file name setting, etc.; (3) the data processing module includes the processing of raw LiDAR data, point cloud registration, single plant extraction, and phenotype extraction.

### 2.2. UGV Platform Hardware Architecture and Control System

The platform is divided into five parts. The hardware structure diagram (Figure 2a,b) and physical diagram (Figure 2c,d) are shown in Figure 2. Various parts (A, B, C, D, and E) in Figure 2b are described below:

(1) Part A represents the UGV body part (size; 2195 mm × 1900 mm × 2065 mm). It is the main structure of the platform and was developed using the French Dassault Systemes (Solidworks 2019 SP5.0 software) for preliminary design. The lowest height of the chassis from the ground is about 1400 mm. It is mainly made of stainless steel and aluminum alloy to reduce weight. The weight of the whole machine is about 200 kg. The tire is made of solid rubber (outer diameter; 660 mm and width; 35 mm). The narrow tire width enables the machine to move flexibly between plants. The UGV is a four-wheel drive machine with a Brushless Direct Current Motor (BLDC) and a rated power and speed of 13.5 kW and 3600 r/min, respectively. The steering motor contains a Brushed Direct Current motor (BDC) with a rated power and speed of 0.95 kW and 1000 r/min, respectively.

(2) Part B represents the electric slide rail installed at the bottom of the chassis. It is connected to the industrial control computer (IPC) through rs485 to the USB. It can control the movement direction, speed, and start/stop position of the slide rail through the control software on the computer.

(3) Part C represents the LiDAR model VLP-16 (Velodyne, CA, USA, Silicon Valley, CA, USA) installed on the electric slide rail. This part contains 16,360-degree scan lines, horizontal measurement angle resolution (0.1° to 0.4°), vertical measurement angle range of 30 degrees, and angle resolution of 2°. The LiDAR is installed on the slide rail at the height of 1 m from the ground. The IPC is connected to the LiDAR through Ethernet, and it can control the LiDAR and collect the data acquired from the LiDAR sensor.

(4) Part D represents the control box and battery module of the moving part of the vehicle. A symmetrical layout with four-wheel drive is used to control the movement of the platform. An independent brushless motor drives each wheel. Meanwhile, four independent motors steer the four wheels. The UGV adopts two steering schemes: four-wheel steering with front and rear wheels in the same direction and front-wheel steering (Figure 3). The steering scheme can be switched using a remote control, depending on the scenario. The turning radius is relatively small when using front and rear-wheel steering and can complete the directional translation in a narrow space. However, the turning radius is larger when using front-wheel steering, and the orientation of the vehicle can be adjusted. The machine is controlled by an Arduino control board, four absolute encoders, and an IPC, which form a closed-loop control system. The IPC can control the steering of the four wheels through the encoder and the Arduino control board. The IPC controls UGV movement by controlling the drive motors of the four wheels. RS485 communication is used among the encoder, driver, Arduino control board, and IPC. The whole vehicle is powered by two lead-acid batteries and a power regulator.

(5) Part E represents the IPC, wireless router, and RTK-GPS module. The UGV platform obtains positioning information using a real-time dynamic differential positioning system (RTK-GPS). A laptop computer connected to the IPC via RDP remote desktop can control the electric slide rail and LiDAR. The laptop can be connected to RTK-GPS using a USB to plan the direction for the UGV. The control architecture diagram of this UGV platform is shown in Figure 4.

### 2.3. Data Acquisition

The designed UGV could be operated manually or by FrSkyTaranisX9DPlus2019 creating tasks from measured GPS points to achieve UGV navigation on a specific path in the field, where centimeter-level GPS information is obtained from RTKGNSS receivers. Therefore, relevant interactive control software that could run on an IPC was developed for the cooperative control of the LiDAR and the electric slide rail. The UGV traverses a strip of potted crops for LiDAR data collection in a fixed-point acquisition manner, i.e., when the UGV stops at a location, the electric slide rail starts to work, carrying the LiDAR from one end of the slide rail to the other at a uniform speed. The speed of the slide rail was set to 344 cm/min based on data from several experiments. A faster speed may lead to the jittering of the slide rail, thus degrading the quality of the point cloud, while a slower speed may increase the amount of data, leading to less efficiency and is also not conducive for the transmission, storage, and subsequent processing of data. The LiDAR starts to record the point cloud in .pcap file format when the slide rail starts moving and stops recording when it reaches the end of the slide, after which the UGV starts to move to the next position for the next round of LiDAR data acquisition. The data acquisition method is shown in Figure 5.

### 2.4. Data Preprocessing

The data acquired by LiDAR is a pcap data package, which is dynamic data composed of 16 lines of laser points at 15 frames per second. The original data of LiDAR must be processed to get the dense 3D point cloud of the plant. The traditional point cloud stitching method was used to estimate the bit attitude of each frame acquired by LiDAR through a wheeled odometer or laser odometer. Wheel odometer may lead to splicing errors because the ground is uneven, and the UGV may have bumps. The robustness of slam building using the laser odometer method is very poor since the plant leaves are flexible and can be easily affected by the environment. In the experimental design, the speed of the slide rail was a mm/s. The LiDAR was used to acquire data at 15 frames per second. Assuming that each point P in the kth frame is being processed, P’s coordinates can be translated along the vector d to obtain the following equation:(1)$$P^{\prime }=P+d$$
where,
(2)$$P=\left(\begin{array}{c} x\\ y\\ z\\ 1 \end{array}\right),P^{\prime }=\left(\begin{array}{c} x^{\prime }\\ y^{\prime }\\ z^{\prime }\\ 1 \end{array}\right),d=\left(\begin{array}{c} \alpha_{x}\\ \alpha_{y}\\ \alpha_{z}\\ 1 \end{array}\right)$$

The transformation process from P to P’ can be expressed as P’=TP, using the following transformation matrix T:(3)$$T=T\left(\alpha_{x},\alpha_{y},\alpha_{z}\right)=\left(\begin{array}{c} 1\\ 0\\ 0\\ 0 \end{array}\;\begin{array}{c} 0\\ 1\\ 0\\ 0 \end{array}\;\begin{array}{c} 0\\ 0\\ 1\\ 0 \end{array}\;\begin{array}{c} \alpha_{x}\\ \alpha_{y}\\ \alpha_{z}\\ 1 \end{array}\right)$$

Since the LiDAR is moving in the *z*-axis direction towards the sensor, Equation (3) can be expressed as follows:$$\alpha_{x}=0,\alpha_{y}=0,\alpha_{y}=\frac{\mathrm{ka}}{15}$$

Each point P of each frame can be derived from the actual spatial coordinates. A Velodyne data processing software was developed accordingly. The method of uniform superposition was used to stitch each frame into a dense point cloud of UGV fixed-point blocks. The Velodyne data processing flow is shown in Figure 6.

### 2.5. Point Cloud Processing and Phenotype Estimation

An automated processing pipeline was developed using some library functions in open3d. Python language, Visual Studio 2019 IDE, and the automated processing pipeline were used to post-process the block point clouds and perform crop phenotype estimation. Firstly, all block point clouds acquired by UGV in the crop strip were spliced, then normalized to remove the noise, and the ground was fitted. Each crop was then segmented using a clustering algorithm, and the phenotypic parameters, including plant height and maximum crown width, were extracted for each crop. The point cloud processing pipeline is shown in Figure 7.

#### 2.5.1. Point Cloud Registration

The block point cloud needs to be stitched after preprocessing in the order acquired by the UGV to form a whole strip of the point cloud. In this paper, the block point cloud was aligned using the Iterative Closest Point (ICP) algorithm [28]. However, ICP has a high requirement on the initial positions of the aligned point cloud and the reference point cloud. The algorithm is prone to local optimum after registration if the initial positions of the two point clouds are very different. Therefore, the two point clouds should be aligned first before using this algorithm for registration. Each block of the data acquired by the UGV has an equal spacing of 190 cm with an overlap of 30% between each adjacent two blocks. Therefore, coarse registration can be completed by moving the kth block 190 (k − 1) cm in the *z*-axis direction. Moreover, the ICP algorithm is based on the least squares method, which finds the nearest neighbors based on certain constraints to calculate the best registration parameters, i.e., the rotation matrix R and the translation vector t, which is the minimum value of the error function. The error function E (R, t) can be expressed as follows:(4)$$E\left(R,t\right)=\frac{1}{n}\sum_{i=1}^{n}\|q_{i}-{\left({\mathrm{Rp}}_{i}+t\right)\|}^{2}$$
where n, qi, R, and t represent the number of nearest neighbor pairs, a point in the target point cloud P, the nearest point in the source point cloud Q corresponding to pi, the rotation matrix, and the translation vector, respectively. The ICP algorithm is implemented as follows:

(1) The set of points pi ∈ P in the target point cloud P is selected;

(2) The set of points qi corresponding to pi in the source point cloud Q is identified (satisfying qi ∈ Q such that is the minimum value);

(3) The rotation matrix R and translation vector t are obtained by calculating the relationship between the corresponding point sets such that the error function is minimized;

(4) pi is transformed, and a new set of corresponding points is obtained;

(5) The average distance d between the new pi and its corresponding point set qi is calculated. The distance d is calculated as follows:(5)$$d=\frac{1}{n}\sum_{i=1}^{n}\|p_{i}-q_{i}\|{}^{2}$$

(6) The calculation is stopped if d is less than the given threshold or greater than the set number of iterations; otherwise, step (2) and the subsequent processes are re-executed until the convergence conditions are met. Herein, the registration of the two point clouds was simultaneously performed (Figure 7b), which were simultaneously added to the global point cloud to obtain a complete point cloud of the crop strip. The complete point cloud of the strip is shown in Figure 7c.

#### 2.5.2. Noise Removal and Ground Detection

Laser scanning produces point cloud datasets with non-uniform density. In addition, errors in the measurements can produce sparse outliers, leading to poor results. In this study, the outliers or coarse points caused by measurement errors were removed using a Statistical Outlier Removal as follows: a statistical analysis was performed on the neighborhood of each point to calculate its average distance to all neighboring points. Points whose mean distance is outside the standard range (defined by the global distance mean and variance) can be defined as outliers and removed from the data if the result obtained is a Gaussian distribution whose shape is determined by the mean and standard deviation. The ground must be detected after noise removal and the ground points removed. In this study, the Random sample consensus (RANSAC) algorithm was used to distinguish the detected ground from the crop point clouds to be clustered and prevent the ground point clouds from being mistakenly detected as crops in the subsequent clustering process. In the algorithm design, the number of iterations of the algorithm, the distance error threshold, and the total number of points of the point cloud were 1, ΔT1, and N, respectively. First, three points were randomly selected to form the ground to be fitted. The 3D coordinates of the three points were set to (X1, Y1, Z1), (X2, Y2, Z2), (X3, Y3, Z3). The fitted plane model is shown below:(6)Ax+Bx+Cz+D=0
where,
A = (Y2 − Y1) (Z3 − Z1) − (Z2 − Z1) (Y3 − Y1)
B = (Z2 − Z1) (X3 − X1) − (X2 − X1) (Z3 − Z1)
C = (X2 − X1) (Y3 − Y1) − (Y2 − Y1) (X3 − X1)
D = −(AX1 + BY1 + CZ1)

The distance L from any point in space (X0, Y0, and Z0) to this plane was calculated as follows:(7)$$L=\frac{\left|{\mathrm{AX}}_{0}+{\mathrm{BY}}_{0}+{\mathrm{CZ}}_{0}+D\right|}{\sqrt{A^{2}+B^{2}+C^{2}}}$$

A point was inside the model when L between a point and this hypothetical plane was ≤ΔT1. The number of interior points of the model was recorded by iterating through N-3 points instead of the initial sampled three points. The number of interior points of the model was obtained in the same way by randomly sampling three points and constructing a planar model (Iterating 1 time according to this random sampling method). The probability of producing a reasonable result increases with an increasing number of iterations. Finally, the ground model with the highest number of interior points was selected as the best fit based on the number of interior points of each model. The ground detection results are shown in Figure 7d. The red points in the figure represent ground points.

#### 2.5.3. Single Plant Division

In this study, partitioning of the crop point cloud for each plant was perfumed using the Euclidean clustering algorithm to group points close to each other into one category. Assuming that there are n points in point cloud C, the Euclidean distance is defined as the closeness of two points. The distance between the nearest neighboring points is used to achieve the point cloud clustering segmentation. The specific segmentation process is as follows: for the preprocessed point cloud data set P, determine a query point Pi, and set the distance threshold r; find the n nearest neighbor points Pj (j = 1,2, ⋯, n) through KD-Tree; calculate the Euclidean distance dj from the n nearest neighbor points to the query point using Equation (8); compare the distance dj with the distance threshold r; put the points less than r into the class M. The segmentation is completed when the number of points in M is not increasing.
(8)$$d\left(p_{i},p_{j}\right)=\sqrt{\sum_{k=1}^{n}{\left(p_{\mathrm{ik}}-p_{\mathrm{jk}}\right)}^{2}}$$

This algorithm can only partition the single crop where there is no overlap of leaves between two pots. Euclidean clustering cannot partition a single crop where some crops have large leaf growth and are close to each other. In such cases, the K-means clustering algorithm was used for partitioning. The K-means algorithm belongs to the division clustering algorithm, where the mean value of all objects in the cluster represents the center of each cluster. Its input is the number of clusters (K) with the data set containing n objects (D), while the output is the set of K clusters.

The algorithm flow is shown below:(1)Choose any K objects from D as the initial clustering centers.(2)Assign each object to the most similar cluster based on the mean value of the objects in the cluster.(3)Update the cluster mean, i.e., the mean of the objects in each cluster is recalculated.(4)Repeat steps (2) and (3) until the clusters do not change.

The K-means clustering method can determine the number of clusters and initial cluster centers in advance. In this study, four crops could not be partitioned by Euclidean clustering (Figure 7e), and thus K-means clustering algorithm was used for partitioning (K = 4). The partitioning results are shown in Figure 7f.

#### 2.5.4. Plant Height Extraction

The direction of the point cloud obtained by LiDAR and the direction of the xyz coordinate axis of the real-world coordinate system are inconsistent. Therefore, the whole point cloud should be calibrated to the horizontal plane before extracting the height of the plant. Furthermore, the approximate ground should be segmented by the RANSAC algorithm before the point cloud processing pipeline for the ground plane equation to be estimated by the segmented ground points). The rotation matrix R can then be found using the normal vector a(a,b,c) of the ground before horizontal calibration and the vector b(0,0,1) of the LiDAR point cloud coordinate system vertically upward. The point cloud after horizontal calibration can be obtained by multiplying the original point cloud by the rotation matrix R. The plant height (h) can be calculated by finding the difference between the z value (Z_max_) of the highest point of the crop and the z value (Z_min_) of the ground plane, then subtracting the known height (h_p_) of the flower pot (Figure 7g). The plant height calculation formula is shown below:(9)$$h=Z_{\max}-Z_{\min}-h_{p}$$

#### 2.5.5. Maximum Crown width Extraction

The extraction of the maximum crown width of a single crop is essentially a search for the farthest point pair in the leaf plane point cloud. The search for the farthest point pair can be performed using geometric properties. First, the projection in the vertical direction of the monocrop point cloud is calculated, i.e., the z-coordinate value of all points in the monocrop point cloud is 0. This changes the monocrop point cloud from a 3D point cloud to a 2D point cloud in the xy-plane. Subsequently, the convex polygon contours of this 2D point cloud can be extracted to obtain the convex packet. The farthest point pairs of convex packets in a planar point cloud can be calculated using the algorithm proposed by Shamos (1978) for calculating n-point convex packet pairs of anti-podal points (rotational hull method) as follows:Calculate the endpoints in the y-direction of the convex polygon (ymin and ymax).Construct two horizontal tangents from ymin and ymax. Calculate the distance between the pairs and maintain it as a current maximum since they are already a pair of anti-podal points.Simultaneously rotate the two lines until one coincides with one of the sides of the polygon.A new pair of anti-podal points is generated at this time. The new distance is calculated and compared with the current maximum and updated if it is greater than the current maximum.Repeat the process in steps 3 and 4 until a pair of the anti-podal points (ymin and ymax) is produced again.Output the pair of anti-podal points determining the maximum diameter.

The time complexity of this algorithm is O(n). A pair of anti-podal points with the maximum diameter calculated using the above algorithm represents the maximum canopy width (L) of this single crop (Figure 7h).

## 3. Materials

The experiments were conducted in a joint greenhouse of the Beijing Academy of Agriculture and Forestry (39°56′ N, 116°16′ E). The lettuce planting area measured 10 m × 40 m. Six lettuce types (C, W, R, S, B, and L representing Crisphead lettuce, Wild lettuce, Romaine, Stem lettuce, Butterhead lettuce, and loose-leaf lettuce) were grown in pots. Three varieties of each type were planted, totaling 18 varieties with 3 replicates for each variety. One lettuce plant was planted per pot. The pots were under normal water and fertilizer management. The data were acquired on 11, 12, and 13 November 2021. We obtained 54-point cloud data of potted plants. A commercial mobile laser 3D plant phenotyping platform PlantEye F500 developed by Phenospex B.V. (Heerlen, the Netherlands), was also used (http://phenospex.com/products/plant-phenotyping/science-planteye-3d-laser-scanner/, accessed on 15 October 2021) to obtain point cloud data. The measurement principle and physical installation of this sensor are shown in Figure 8. Furthermore, the plant height and maximum crown width were extracted from the obtained point cloud data and compared with the manual measurements through the point cloud processing and analysis pipeline. The extraction results are shown in Table 1. The acquired lettuce strips were arranged according to the width of the UGV since the width of the UGV was fixed.

## 4. Results

### 4.1. Point Cloud Quality

The point cloud data of the potted lettuce were obtained using the UGV-LiDAR phenotyping platform and PlantEye F500. The single point cloud of each lettuce plant was segmented through the developed point cloud processing pipeline. The visualization of the single-point cloud of the lettuce plants obtained by the two platforms is shown in Figure 9.

Although the lettuce point cloud obtained by PlantEye F500 showed a better view due to the RGB information, a part of the leaf may be missing in reality. The obtained leaf point cloud only had the points of the upper surface leaves with only a thin layer and did not provide information on the lower part of the obscured leaves. The point cloud of single lettuce obtained by the designed phenotype platform contained the echo intensity information. The green shade in the figure represents the laser echo intensity. However, this method did not provide RGB information. Nevertheless, the platform obtained more leaf points than the leaf points obtained by PlantEye F500, and contained the lettuce leaf points that are not severely obscured. The thickness of individual leaves obtained by the platform also increased compared with the thickness obtained by PlantEye F500. PlantEye F500 uses a low-powered single-line laser with weak penetration and thus can only detect the upper leaves. Meanwhile, the UGV-LiDAR phenotyping platform uses a more powerful 16-line LiDAR and can handle multiple echoes, and thus can detect the obscured leaves. It can also detect the reflected light from the laser within a single leaf, thus increasing the thickness of the single leaf point cloud. Therefore, the point cloud of a single lettuce plant obtained by UGV-LiDAR phenotyping platform had better point cloud integrity than that obtained by PlantEye F500, which has some advantages in the phenotype detection of canopy 3D structure outer contour of crops.

### 4.2. Evaluation of Phenotypic Parameter Accuracy

The lettuce plant height and maximum crown width obtained by the UGV-LiDAR phenotyping platform and PlantEye F500 were extracted through the lettuce point cloud processing and analysis pipeline. This paper used the coefficient of determination (R2) and root mean square error (RMSE) to evaluate the degree of agreement with the manually measured values. The results showed that the accuracy of the point cloudplant height estimation obtained by the UGV-LiDAR phenotyping system designed in this study is good (R2: 0.97996 and RMSE: 1.51 cm) (Figure 10). PlantEye had a poorer estimation accuracy of the point cloud plant height than UGV-LiDAR (R2: 0.93751 and RMSE: 2.54 cm). Furthermore, the estimation accuracy of the point cloud maximum canopy width obtained by UGV-LiDAR was poorer than that of plant height. Nevertheless, the accuracy of the point cloud maximum canopy width estimation obtained by PlantEye (R2: 0.91798 and RMSE: 5.25 cm) was similar to that of the UGV-LiDAR phenotyping system (R2: 0.90975 and RMSE: 4.99 cm). Correlation analysis showed that both systems could accurately measure the plant height and maximum canopy width of the vegetables. However, the developed UGV-LiDAR phenotyping system could estimate plant height and maximum crown width more accurately than the PlantyEye F500 system.

### 4.3. Performance and Cost

PlantEye F500 is a single-line laser mounted on the orbital overhead travel phenotyping platform and was established in the greenhouse of Beijing Academy of Agriculture and Forestry. The moving speed of the orbit was set to 300 cm/min in actual use. The orbit can acquire about 1020 plants per hour without stopping. Although the moving speed of Velodyne VLP-16 LiDAR on the vehicle track was faster (344 cm/min), it had a slower acquisition speed (810 plants per hour) than PlantEye F500 because the vehicle requires time to move. The PlantEye F500 costs about $147,000, while the UGV-LiDAR phenotyping platform costs only $11,780, which is significantly lower. The point cloud pipelining process is run on a desktop workstation (Intel Core i7 processor, 2.9 GHz CPU, 32 GB RAM, Windows 11 OS). Moreover, the point cloud of lettuce acquired by the UGV-LiDAR phenotyping system was more dense, with an average of about 380,000 points per plant and a processing time of 12,628 ms, while the point cloud acquired by PlantEye F500 was sparse, with an average of about 100,000 points per plant and a processing time of 3157 ms. This indicates that the UGV-LiDAR phenotyping system was less efficient in post-processing than PlantEye F500. A comparison of performance and cost is shown in Table 1.

### 4.4. Morphological Differences between Different Categories of Lettuce

The lettuce phenotypic parameters obtained from the UGV-LiDAR phenotyping platform can be used to determine the differences in morphological traits among various lettuce types. Herein, the mean and variance of lettuce plant height and maximum crown width obtained from the UGV-LiDAR phenotyping platform were calculated using the software SPSS 25.0. A statistical analysis of significant differences was also performed (Figure 11). At least one lettuce variety had significantly different plant height and maximum crown width from the other varieties (Kruskal–Wallis test and Mann–Whitney U test, *p* < 0.05). The Stem lettuce had the highest mean plant height and the largest mean maximum crown width, while Butterhead lettuce had the shortest and smallest mean maximum crown width. The plant height of Stem lettuce was significantly different from that of Crisphead lettuce, Butterhead Lettuce (*p* < 0.01), and Loose-leaf lettuce (*p* < 0.05). Furthermore, the plant height of Butterhead lettuce was significantly different from that of Wild lettuce (*p* < 0.01) and Romaine (*p* < 0.05). The maximum crown width of Stem lettuce was significantly different from that of Butterhead lettuce (*p* < 0.01) and Crisphead lettuce (*p* < 0.05). Moreover, the maximum crown width of Butterhead Lettuce was significantly different from that of Wild lettuce and Romaine (*p* < 0.05). In conclusion, the surfaces of the UGV-LiDAR phenotyping platform and 3D point cloud resolution pipeline are sensitive enough to detect subtle differences between different lettuce types.

## 5. Discussion

The comparison results showed that the phenotypic platform and phenotypic parameter extraction pipeline could reliably measure the plant height and maximum crown width of greenhouse-potted crops using the synergistic operation of LiDAR and track, thus helping breeders to easily observe and screen good traits in many samples. PlantEye F500 is a well-established commercial plant 3D scanner used for automatic and continuous observation of plant growth status [26]. Compared with PlantEye F500, the UGV had higher estimation accuracy for plant height and lower estimation accuracy for maximum crown width. Moreover, UGV was less costly compared with other phenotyping platforms, such as the suspension phenotyping platform [11], orbital overhead travel phenotyping platforms [8], and other immovable on-site phenotypic platforms. Furthermore, these platforms are difficult to disassemble and install to other plots after construction is completed, while UGV and UAV platforms can be used in any plot due to their high flexibility [29]. However, the turbulence caused by the rotor blades of UAVs in low flight may significantly affect the plant canopy structure, leading to a large error when measuring phenological parameters. Moreover, the resolution obtained by the sensors is usually very low when the UAVs reach a height where the airflow does not cause disturbance to the plants. The resolution of the images or point clouds obtained by UGV may be higher than that obtained by the UAV since the sensors on the UGV phenotyping platform are closer to the top of the plant canopy.

However, UGVs have some disadvantages. First, the quality of the ground soil limits UGV movement. For example, wet soil will make the UGV stuck into the mud, leading to the compaction of the soil and damage to plants. The traditional UGV phenotype platform continuously moves while collecting data. The movement bumps can affect point cloud acquisition. Therefore, the LiDAR frames should be stitched to obtain the complete plant canopy 3D morphology for a high-density point cloud. The jitter of the vehicle may also lead to difficulties in using wheel odometry, laser odometry, and other ways of SLAM build map. If the LiDAR frames are not stitched, a high line number of LiDAR would be needed to obtain a dense point cloud [30]. However, the increased number of LiDAR lines increases the cost. Furthermore, the maximum canopy estimation accuracy may not be as high as the plant height estimation accuracy due to the difficulty of manually measuring the maximum canopy width. Unlike the UGV, which causes contact interference with larger plants during movement, the new UGV-LiDAR phenotyping platform has a small electric slide rail for efficient data acquisition while the plants are stationary, leading to high accuracy. Meanwhile, the LiDAR sensors are closer to the plants, increasing accuracy. The small slide rail also increases the running accuracy compared with the track of the orbital overhead travel phenotyping platform.

However, the proposed UGV-LiDAR phenotyping system has some limitations and unfinished parts: (1) the point cloud data acquired by the UGV-LiDAR phenotyping system has a large amount of redundancy, leading to inefficiency in post-processing. A denser point cloud is unsuitable in acquiring these phenotypic parameters of plant height and maximum crown width since it represents more noise, which affects the extraction of phenotypic data. (2) Subsequent studies should optimize the UGV-LiDAR phenotyping system using faster and more accurate vehicle-mounted motorized slide rail and replacing LiDAR with higher accuracy and lower line count. Further studies should increase resolvable phenotypic parameters, such as leaf number, width, inclination, area, etc. Besides, deep learning should be used to improve the speed and accuracy of lettuce single plant identification. (3) Future studies should design UGV with adjustable height and span for application to more plants, more scenes, and the acquisition of phenotypic information of plants during different growth and development periods. (4) Finally, various sensors, such as RGB camera, thermal infrared camera, and multispectral camera should be used in the future to monitor more phenotypic information of plants.

## 6. Conclusions

In this paper, a new UGV phenotype platform equipped with an electric slide rail and phenotype data acquisition-analysis pipeline was proposed to avoid the effect of movement bumps on the quality of point cloud acquisition. The platform was developed using 16-line LiDAR, electric slide rail, and UGV via RTK-GPS for automatic movement to obtain fine point cloud data. The 3D structure of the lettuce canopy was obtained by the homogeneous overlay frame method based on uniform speed superposition frames to lettuce. This method has a cost advantage compared with the traditional UGV high line count LiDAR point cloud acquisition system. The point cloud was matched and fused by the iterative nearest point (ICP) algorithm through pipelining to complete the 3D reconstruction of a whole strip point cloud. Random Sampling Consistency (RANSAC) algorithm, Euclidean clustering, and k-means clustering algorithm were used to obtain a single lettuce canopy 3D point cloud. The plant height and maximum crown width were also accurately estimated. The new UGV phenotype platform can be used to accurately measure plant height and maximum crown width with high accuracy and at a reduced cost compared with PlantEye F500. Therefore, the platform can be used to measure other plant 3D phenotype data after further expansion of the algorithm. The UGV platform can also be installed with other sensors to achieve more dimensional phenotype information monitoring.

## Figures and Tables

**Figure 1 plants-12-00483-f001:**
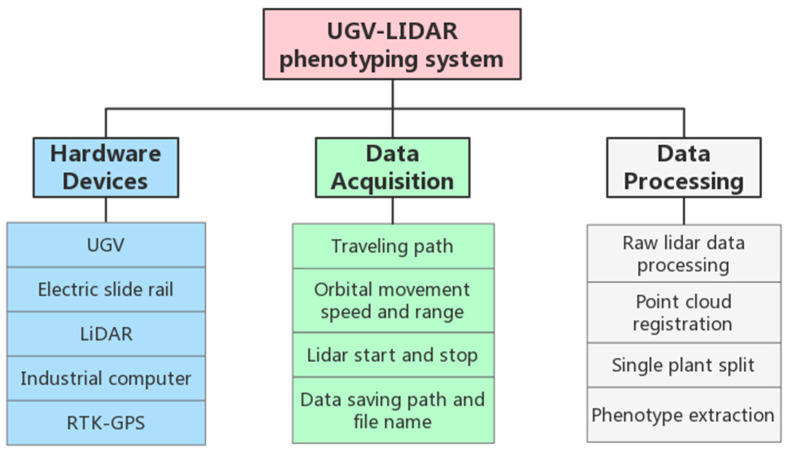
Composition of UGV-LiDAR phenotyping system.

**Figure 2 plants-12-00483-f002:**
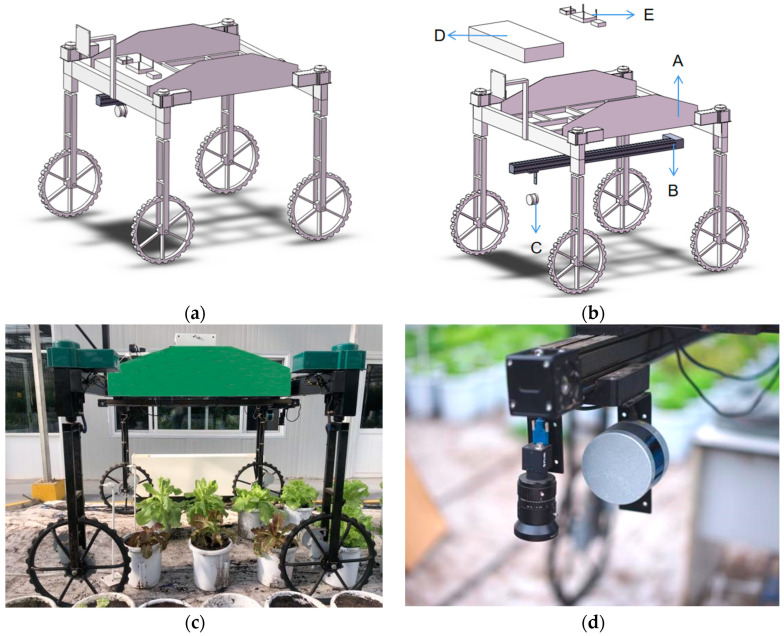
Hardware structure and physical structure of the UGV. (**a**) Three-dimensional view; (**b**) decomposition diagram; (**c**) real view of the UGV collecting data; (**d**) LiDAR installation diagram.

**Figure 3 plants-12-00483-f003:**
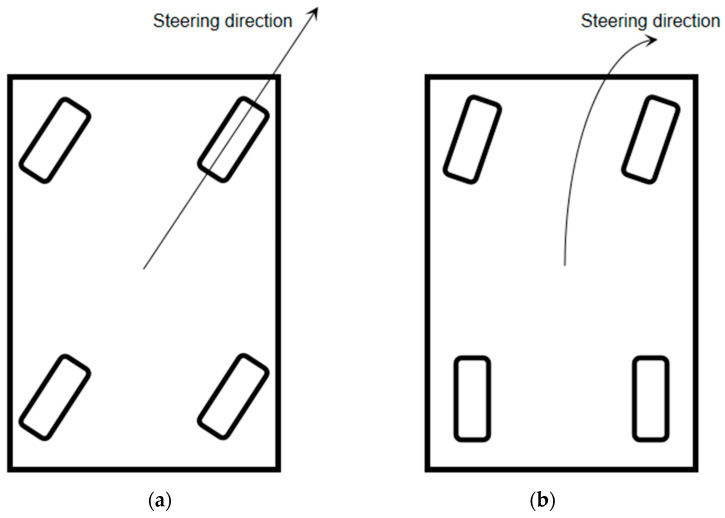
Schematic diagram of two steering schemes. (**a**) Four-wheel co-steering; (**b**) front-wheel steering.

**Figure 4 plants-12-00483-f004:**
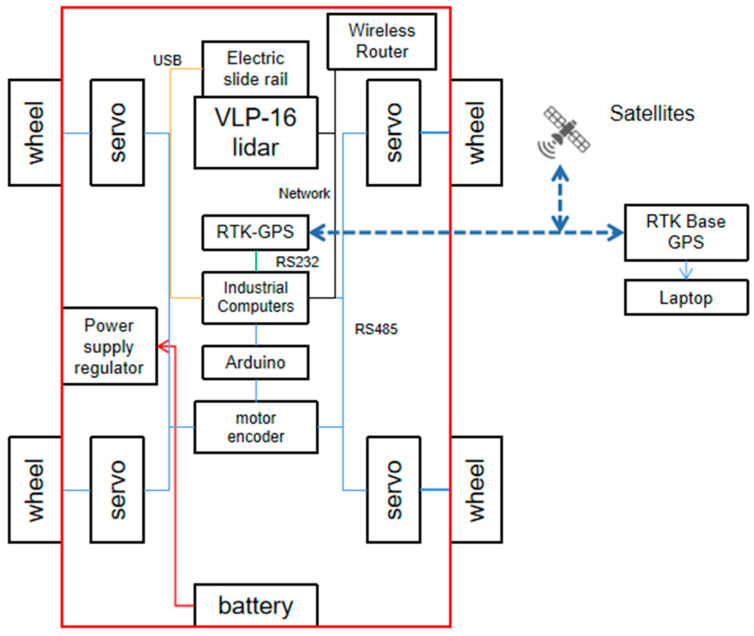
UGV control system.

**Figure 5 plants-12-00483-f005:**
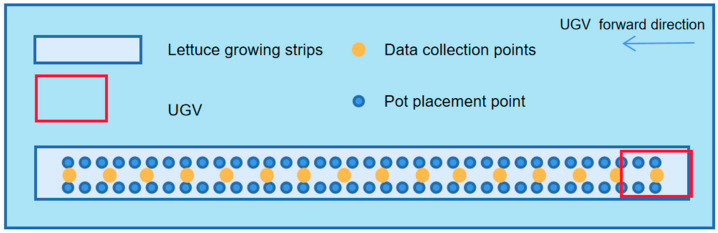
Schematic of UGV-LiDAR phenotyping platform for data acquisition.

**Figure 6 plants-12-00483-f006:**
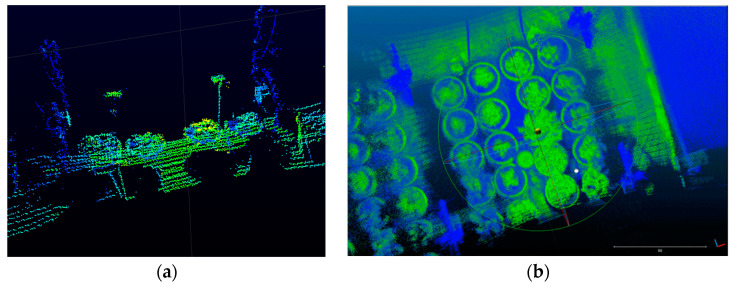
Velodyne data processing flow. (**a**) Velodyne raw data. (**b**) Dense point cloud of UGV fixed-point block.

**Figure 7 plants-12-00483-f007:**
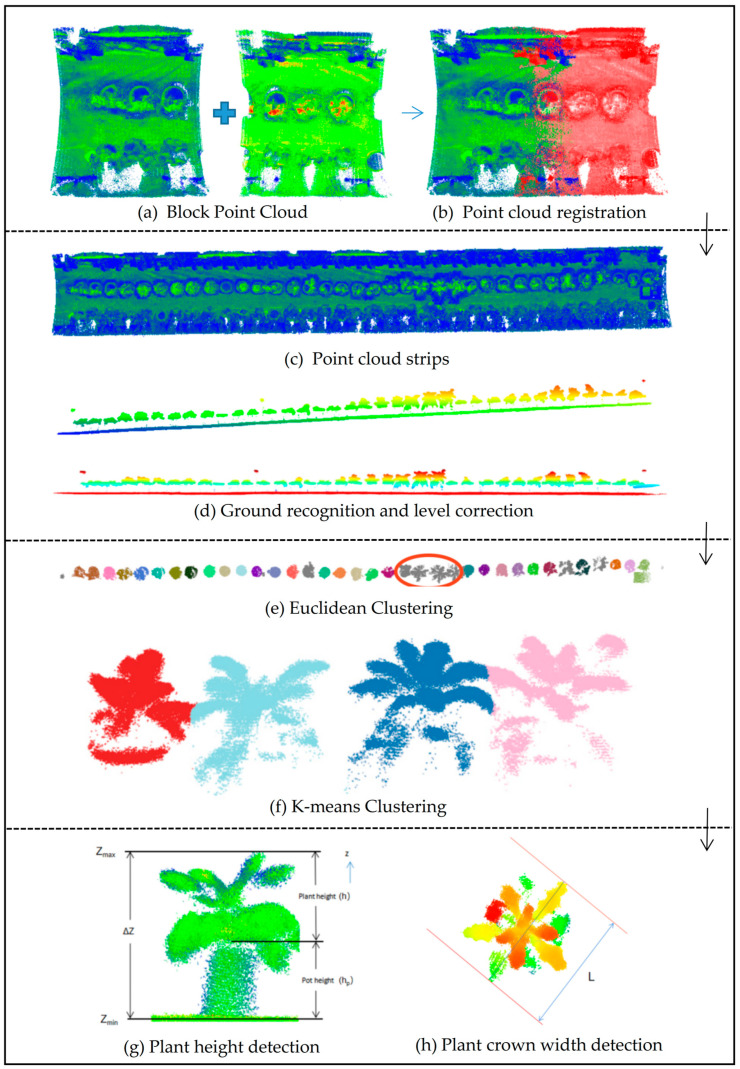
Point cloud processing pipeline.

**Figure 8 plants-12-00483-f008:**
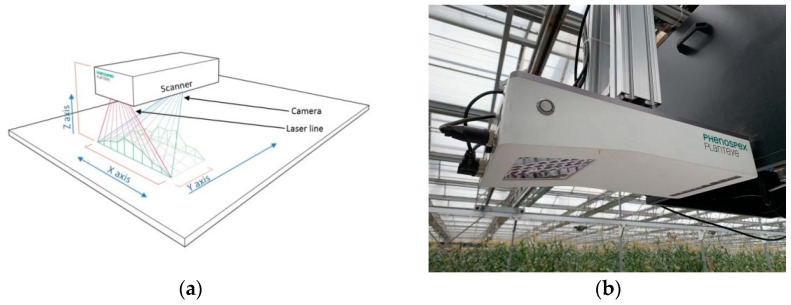
PlantEye F500 measurement principle and physical installation diagram. (**a**) The 3D laser scan sensor. The red and blue lines represent the laser line and reflection of the laser after projecting to the plant to be received by the cmos, respectively. (**b**) The sensor mounted on the overhead orbital system for easy movement.

**Figure 9 plants-12-00483-f009:**
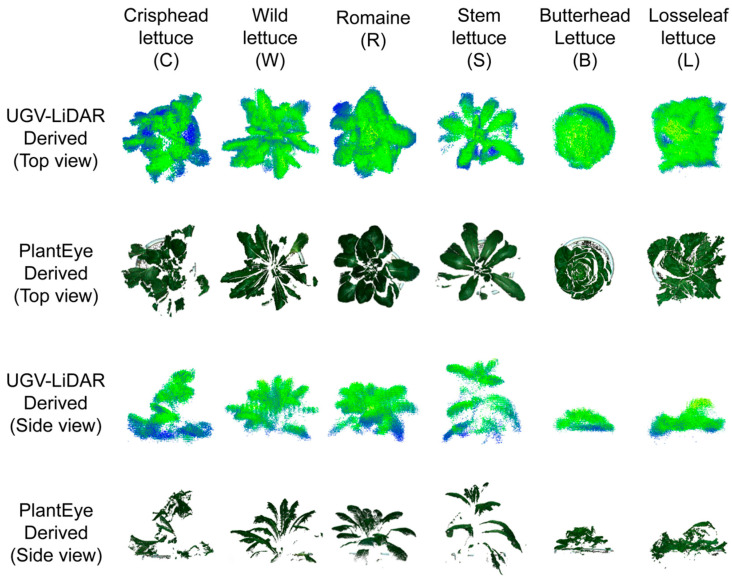
Comparison of the single-point cloud of lettuce plants acquired by UGV-LiDAR platform and PlantEye.

**Figure 10 plants-12-00483-f010:**
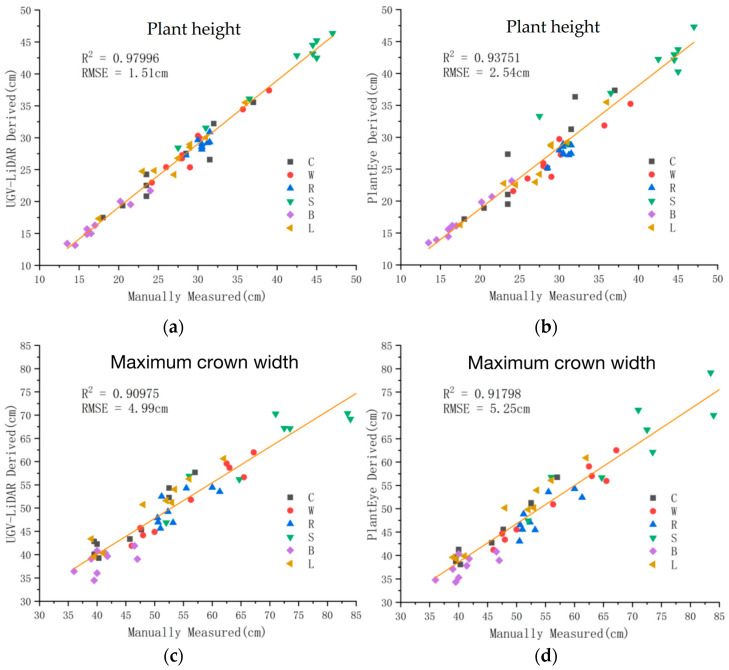
Comparison of plant height and maximum crown width extracted using the point cloud processing pipeline. (**a**) Linear fit of plant height estimated by point cloud obtained by UGV-LiDAR phenotyping system against manual measurements. (**b**) Linear fit of maximum crown width estimated by point cloud obtained by UGV-LiDAR phenotyping system against manual measurements. (**c**) Plant height estimated by point cloud obtained by PlantEye against manual measurements. (**d**) Linear fit of the maximum canopy width estimated by the point cloud obtained by PlantEye against manual measurements.

**Figure 11 plants-12-00483-f011:**
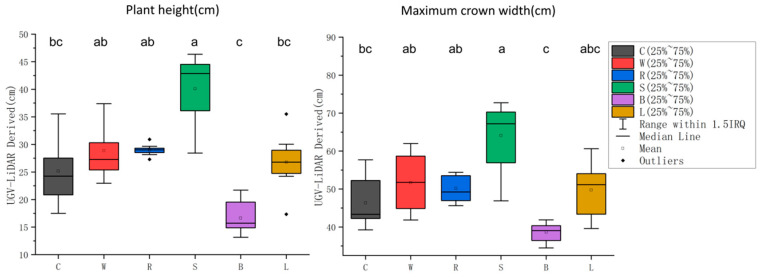
Analysis of plant height and maximum crown width differences. The central horizontal line indicates the median. The top and bottom of the box indicate the 25th and 75th percentile, respectively. The upper and lower solid dots indicate outliers beyond the upper and lower quartiles, respectively; whiskers extend to the extreme non-outliers. The hollow dots represent the mean. Different letters indicate statistically significant differences between species (*p* < 0.05).

**Table 1 plants-12-00483-t001:** Performance and cost comparison between UGV-LiDAR phenotyping system and PlantEye F500.

Parameter Name	UGV-LiDAR Phenotyping System	PlantEye F500
Flux	810 plants/h	1020 plants/h
Cost	$11,780	$147,000
Point cloud density	380,000 points/plant	100,000 points/plant
Pipelining processing time	12,628 ms/plant	3157 ms/plant

## Data Availability

Not applicable.

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
