# Peer review of "Design and Development of a Low-Cost UGV 3D Phenotyping Platform with Integrated LiDAR and Electric Slide Rail"

_plants, 2023, doi:10.3390/plants12030483_

Round 1
Reviewer 1 Report
This research paper (plants-2120512) is well presented. I like this manuscript, in special the cost in relation other equipment and very interesting approach combining different technologies for classification-predicted lettuce plants. My only comment for this research is that the inor check speling english in some sentences. The article is well described (introduction, material and methods, the results and discussion) and properly structured. In addition, Alphabetic order in keywords;
Best regards
Reviewer 2 Report
The article describes the developed inexpensive system for 3D plant phenotyping. The article is well structured and relevant.
There are a few remarks:
Lines 109-111 should be deleted.
Add an abbreviation explanation after the first mention, for example RTK-GPS line 104. Check throughout the text.
Line 132 - indicate the company and country of manufacture. Also for other items, indicate the company and country of manufacture. Check throughout the text.
Figure 7g increase the font height. In my opinion, it is worth adding explanations to the figure in the text.
You should add a subsection - statistics.
